# Risk Factors in HIV-1 Positive Patients on the Intensive Care Unit: A Single Center Experience from a Tertiary Care Hospital

**DOI:** 10.3390/v15051164

**Published:** 2023-05-13

**Authors:** Arik Bernard Schulze, Michael Mohr, Jan Sackarnd, Lars Henning Schmidt, Phil-Robin Tepasse, Felix Rosenow, Georg Evers

**Affiliations:** 1Department of Medicine A, Hematology, Oncology and Pulmonary Medicine, University Hospital Münster, 48149 Münster, Germany; 2Department of Cardiovascular Medicine, Internal Intensive Care Medicine, University Hospital Münster, 48149 Münster, Germany; 3Medical Department IV, Pneumology, Respiratory Medicine and Thoracic Oncology, Klinikum Ingolstadt, 85049 Ingolstadt, Germany; 4Department of Internal Medicine II, University Hospital Regensburg, 93053 Regensburg, Germany; 5Department of Medicine B, Gastroenterology, Hepatology, Endocrinology and Clinical Infectiology, University Hospital Münster, 48149 Münster, Germany

**Keywords:** HIV, ICU, SAPS 2, SOFA, APACHE II

## Abstract

HIV-positive patients with acquired immunodeficiency syndrome (AIDS) often require treatment on intensive care units (ICUs). We aimed to present data from a German, low-incidence region cohort, and subsequently evaluate factors measured during the first 24 h of ICU stay to predict short- and long-term survival, and compare with data from high-incidence regions. We documented 62 patient courses between 2009 and 2019, treated on a non-operative ICU of a tertiary care hospital, mostly due to respiratory deterioration and co-infections. Of these, 54 patients required ventilatory support within the first 24 h with either nasal cannula/mask (n = 12), non-invasive ventilation (n = 16), or invasive ventilation (n = 26). Overall survival at day 30 was 77.4%. While ventilatory parameters (all *p* < 0.05), pH level (c/o 7.31, *p* = 0.001), and platelet count (c/o 164,000/µL, *p* = 0.002) were significant univariate predictors of 30-day and 60-day survival, different ICU scoring systems, such as SOFA score, APACHE II, and SAPS 2 predicted overall survival (all *p* < 0.001). Next to the presence or history of solid neoplasia (*p* = 0.026), platelet count (HR 6.7 for <164,000/µL, *p* = 0.020) and pH level (HR 5.8 for <7.31, *p* = 0.009) remained independently associated with 30-day and 60-day survival in multivariable Cox regression. However, ventilation parameters did not predict survival multivariably.

## 1. Introduction

*Human immunodeficiency virus 1* (HIV-1) is a human pathogen retrovirus [1] that primarily infects a cluster of differentiation 4 positive (CD4^+^) T-helper-cells, macrophages, and dendritic cells [2]. In addition, interaction of HIV-1 with erythropoiesis and megakaryopoiesis and concomitant anemia and thrombocytopenia have been described [1,2,3,4,5]. After infection, HIV-1 continuously damages the hosts cellular immune response, facilitates opportunistic infections, and also predisposes to the development of immunologic and malignant diseases, that might progress into life-threatening acquired immunodeficiency syndrome (AIDS), if untreated.

In 2020, 37.7 million people worldwide were HIV-positive. Fortunately, new HIV infection rates diminished from 2.5 million cases per year in 2004 to 1.5 million cases per year in 2020. In parallel, AIDS-related deaths declined from about 1.75 million cases per year in 2004 to 680,000 per year in 2020 [6,7]. In addition to education, safer sex campaigns, enforcement of gender equality, destigmatization of HIV-infected people, and, most importantly, a wider accessibility to antiretroviral treatment (ART) reduced the incidence of AIDS-defining courses after HIV infection. However, worldwide only 27,500,000 (i.e., 73%) of HIV-positive people had access to ART in 2020 [6,7], with highly relevant local variations.

While sub-Saharan countries, such as South Africa (i.e., 18.0% prevalence in adults), face the challenge of providing care to 7.6 million HIV-positive people [8,9], prevalence of HIV-positive people in Western Europe, e.g., in Germany, is constantly at about 0.1% of the population [6,10]. In parallel, overall ART coverage in South Africa is only 72% compared to 86% in Germany [6].

Still, only early diagnosis and sustained treatment with ART may convert HIV-1 infection into a chronic disease [11], and, thus, prevent the onset of AIDS as HIV-1 infection’s final stage [12].

AIDS in adults is defined by PCR-confirmed HIV-positivity plus World Health Organization (WHO) stage IV disease (i.e., wasting syndrome, Pneumocystis pneumonia, recurrent severe bacterial pneumonia, chronic herpes simplex infection, esophageal or bronchial candidiasis, Kaposi’s sarcoma, *Cytomegalovirus* infection, central nervous system toxoplasmosis, chronic cryptosporidiosis or isosporiasis, extrapulmonary cryptococcosis, disseminated endemic mycosis [coccidiomycosis or histoplasmosis] or non-tuberculous mycobacterial infection, extrapulmonary tuberculosis, HIV encephalopathy, cerebral B-Non-Hodgkin-Lymphoma, progressive multifocal leukoencephalopathy, symptomatic HIV-associated nephropathy, or cardiomyopathy) or immunological diagnosis with HIV-infection or first documented CD4^+^ cell count < 200/µL [13]. Many of these conditions lead to rapid deterioration of health and might, therefore, require treatment in an intensive care unit (ICU) setting. 

As a consequence of regional variability in epidemiology, absolute numbers of patients treated for AIDS-related complications differ between high-incidence countries, such as South Africa, and low-incidence countries, such as Germany [14]. Nevertheless, ICU staff worldwide is thought to face similar challenges in treatment of this vulnerable predefined population. Regarding this, we examined co-infections, comorbidities, established scoring systems and other potential risk factors in HIV-positive patients treated in non-operative ICU at a tertiary referral hospital to compare with cases reported from countries of different regional incidence and, subsequently, identify early risk factors, that might be used for an individualized prognostic assessment.

## 2. Materials and Methods

### 2.1. General Remarks and Enrollment Criteria

The study was approved by the local ethics committee (Ref. No. 2020-128-f-S). Considering International Classification of Diseases (ICD)-10 codes R75 (i.e., positive laboratory screening test on HIV), B20 (i.e., infectious and parasitic diseases resulting from HIV disease), B21 (i.e., Malignant neoplasms secondary to HIV disease), B22 (i.e., other specified diseases related to HIV disease (e.g., dementia, encephalopathy, interstitial lymphoid pneumonia, cachexia, wasting syndrome)), B23 (i.e., other medical conditions related to HIV disease (e.g., acute HIV-infection syndrome)), and B24 (i.e., unspecified HIV disease), patients treated between 2009 and 2019 were retrospectively identified via database query of the ICU hospital information system (GE Healthcare, Quantitative Sentinel^®^, Barrington, IL, USA) and general hospital information system (Dedalus HealthCare, ORBIS^®^, Bonn, Germany). Exclusion criteria were an ICU-stay less than 24 h, need for observation during procedures performed under short anesthesia (e.g., electro cardioversion, endoscopy with high periinterventional risks, or external pacemaker therapy before operation), patients under 18 years of age, and false-positive HIV-screening tests.

### 2.2. Description of the Intensive Care Unit

All patients included in this analysis were treated in the ICU of the tertiary care university hospital of Münster, Germany. The internal medicine ICU is equipped with 24 beds for critically ill patients. Of these, 12 beds are accessible via airlock sluices to allow treatment of highly contagious patients (e.g., tuberculosis). Each of the 24 beds is equipped with invasive and non-invasive ventilation techniques, on-site hemodialysis (e.g., continuous veno-venous hemodialysis (CVVH), slow low-efficient daily dialysis (SLEDD)), extracorporeal liver support (e.g., MARS^®^), extracorporeal membrane oxygenation (e.g., veno-arterial (va-ECMO), veno-venous (vv-ECMO)), or plasmapheresis and leukapheresis, if needed. Bedside diagnostics include abdominal and thoracic ultrasound and puncture, echocardiography, diagnostic and interventional bronchoscopy, gastroscopy, and colonoscopy, as well as bone marrow and lumbar puncture.

Patient care is provided by a recognized multidisciplinary team of intensive care physicians and critical care nurses, and supported by trained cardiologists, infectiologists, nephrologists, pulmonologists, hematologists, and oncologists. Anti-infective treatments are regularly reviewed by antibiotic stewardship experts. Moreover, the internal medicine ICU is certified as a Cardiac Arrest Center, providing intensive patient care at the highest level.

Annually, more than 2000 patients are treated in this maximum-care intensive care unit in the northwestern German major city of Münster, with a population of 317,713 inhabitants (as of 2021). The catchment area comprises about 6,000,000 people, most of whom live in northern Westphalia, southern Lower Saxony, as well as the eastern part of the Netherlands. Critically ill patients can be admitted to the ICU by rescue helicopter, mobile intensive care unit or ambulance. The central and peripheral laboratory facilities implemented quality management in accordance with DIN EN ISO 15189 standard and ever since have been repeatedly accredited by German accreditation agency DAkkS (D-ML-1302-01-00).

### 2.3. Database Construction

Following pseudonymization, we collected clinical and laboratory data. Next to gender, which, in our cohort, did not differ from biological sex, we documented age, duration of in-hospital stay, duration of ICU stay, cause of ICU admission, HIV-copy count (thousand copies/µL), and CD4^+^ cell count (cells/µL) at ICU admission. In addition, we recorded the date of first HIV diagnosis and initiation of ART regimen. Next, cause of ICU admission, obtained co-infections and specific treatment regimen were gathered. 

Bacterial and fungal co-infections were detected using microbiological cultures of patient samples (i.e., blood cultures, respiratory secretion cultures, stool cultures, and cerebrospinal fluid cultures). If suspected, detection of *Clostridium difficile* toxin via stool specimen was considered sufficient for diagnosis with or without microbial culture [15]. *Pneumocystis jirovecii* pneumonia was regularly diagnosed by polymerase chain reaction (PCR) from bronchoalveolar lavage (BAL) fluid in accordance with clinics and radiography, if Giemsa stain was inconclusive beforehand [16,17]. Except for serological antibody and antigen-derived diagnosis of Hepatitis B [18], other viral infections or reactivations, such as Hepatitis C (i.e., *Herpacivirus*) [19], Epstein–Barr virus (EBV), or *Cytomegalovirus* (CMV), required virus detection via PCR [20,21] performed on blood, cerebrospinal fluid, BAL, smear, stool, or tissue samples, regardless of prior serological antibody testing. However, due to the retrospective design, we could not consistently distinguish between viral reactivation or primary infection and between co-infection with positive laboratory results or organ-threatening manifest disease, e.g., in CMV- or EBV-associated disease.

Following documentation of infectious complications, we coded pre-existing comorbidities and substance abuse. Next, the patient’s height (in centimeter (cm)), weight (in kilogram (kg)), and body temperature (in degrees Celsius (°C)), awareness (measured by Glasgow Coma Scale (GCS) [22,23]), left ventricular ejection fraction and chest X-ray status at ICU admission were recorded. Additionally, the use of renal replacement therapy, the use of extracorporeal membrane oxygenation (ECMO) and prone positioning during ICU stay were documented. Subsequently, the best and worst blood pressure parameters (systolic, mean, diastolic), use of vasopressors (yes/no), heart rate (minimum, maximum), urine volume (in milliliter (mL)), maximum respiratory rate (per minute), ventilation status (worst during first 24 h: invasive ventilation, non-invasive ventilation including high-flow oxygen supplementation, and spontaneous breathing, including nasal cannula and face mask flow), maximum ventilation driving pressure (Pmax in cm H_2_O), on maximum positive end expiratory pressure (PEEP in cm H_2_O) and on fraction of inspired oxygen (FiO_2_ in%; approximated for nasal cannula flow as proposed by Parke et al. [24]: 1 L O_2_/min = 24%, 2 L O_2_/min = 28%, 3 L O_2_/min = 32%, 4 L O_2_/min = 36%, 5 L O_2_/min = 40%, 6 L O_2_/min = 44%) during the first 24 h of ICU stay were collected. Moreover, data on white blood count (/µL), hemoglobin level (in gram/deciliter (g/dL)), hematocrit (in%), platelet count (in thousands/µL), sodium and potassium levels (in mmol/L), arterial oxygen partial pressure (paO_2_ in millimeter mercury column (mmHg)), arterial carbon dioxide partial pressure (paCO_2_ in mmHg), pH, bicarbonate level (HCO_3_^−^ in mmHg), glucose (in milligram/deciliter (mg/dL)) and lactate (in mmol/L) were recorded from blood gas analyses during first 24 h of ICU stay. Additionally, creatinine and bilirubin (in mg/dL), albumin (in g/dL), c-reactive protein (in mg/dL) and—if available—procalcitonin (in nanogram/milliliter (ng/mL)), lactate dehydrogenase (in units/liter (U/L)), and coagulation parameters (i.e., international normalized ratio, Quick value, and the partial thromboplastin time) were gathered. Finally, we assessed survival status and documented the last date of contact. 

Using these data, the Simplified Acute Physiology Score 2 (SAPS 2) (including age, lowest heart rate in 24 h, worst systolic blood pressure, highest body temperature, worst paO_2_/FiO_2_ ratio, total urinary output in 24 h, serum urea levels, white blood cell count, serum potassium level, serum sodium level, serum HCO_3_^−^ level, bilirubin level, Glasgow Coma Scale score, type of admission (i.e., unscheduled surgical vs. scheduled surgical vs. medical), presence of AIDS defining conditions, hematologic malignancy, and metastatic cancer) [25], the Acute Physiology and Chronic Health Evaluation II (APACHE II) Score (including rectal body temperature, mean arterial pressure, heart rate, respiratory rate, oxygenation measured by arterio-alveolar oxygen partial pressure difference (AaDO_2_) if FiO_2_ is >0.5 or measured by paO_2_ if FiO_2_ is <0.5, arterial pH level, serum sodium level, serum potassium level, serum creatinine, hematocrit, white blood cell count, Glasgow Coma Scale score, serum HCO_3_^−^ level, and added up by age, as well as chronic health points differing in their range for elective post-operative patients vs. non-operative or emergency operative patients) [26], the Sequential Organ Failure Assessment (SOFA) Score (including respiratory parameters, platelets, bilirubin levels, reduced mean arterial pressure and/or catecholaminergic support, consciousness measured by Glasgow Coma Scale score, and creatinine levels or urine output) [27,28,29], as well as the Quick SOFA (qSOFA) Score (including respiratory rate, Glasgow Coma Scale score and systolic blood pressure) [28,30] were calculated for each patient during first 24 h of ICU stay. These organ-failure scores (i.e., SAPS2, APACHE II, SOFA and qSOFA) were used to determine a measurable extent of critical condition of the ICU-treated patient.

### 2.4. Statistical Analysis

Data collection, diagramming as well as calculations, were performed using IBM^®^ SPSS^®^ Statistics Version 27 (released 2020, IBM Corp., Armonk, NY, USA). To present generalizable data without the pertinent impact of outliers, we described the cohort by use of raw count and frequencies, as well as mean, standard deviation (SD), median, and 95 percent confidence interval. Twofold associations between categorical variables were analyzed via Fisher’s exact test or Chi squared test, if applicable. Continuous and ordinal variables were tested using either unpaired *t*-test or Mann–Whitney-U test depending on the normality of the data.

The area under the curve (AUC) receiver operating characteristics curve (ROC) was used to approximate optimal cut-off values for laboratory parameters and ICU scores.

Correlations between continuous variables were calculated via the Spearman–Rho test.

The overall survival (OS) included the time (days) between ICU admission and death or censoring. The 30-day survival included the time (days) between ICU admission and death, or censoring before or at day 30. The 60-day survival included the time (days) between ICU admission and death or censoring before or at day 60. Univariate survival analyses compared OS between groups by using Log rank tests. Multivariate survival analyses used Cox regression model. Here, *p*-values were calculated by forward likelihood ratio test with 0.05 inclusion criterion and 0.10 exclusion criterion. The local significance level was set to 0.05. An adjustment to multiple testing was not determined.

## 3. Results

Between 2009 and 2019, we identified 62 patients with HIV-1 infection, who were treated on a non-operative ICU at Münster university hospital. Baseline characteristics of the cohort are shown in Table 1. The cohort included 77% male patients with a mean age of 46.3 years. In 25 cases, HIV diagnosis was made at referral to ICU. Of the other 37 patients with an existing HIV diagnosis, one patient refused ART and eight patients were incompliant on ART. The ART medication used in each case is listed in Appendix A. The average CD4^+^ T-cell count was less than 200/µL and mean HIV-copy burden at ICU admission more than 500,000/µL. The average duration on ICU was 14.48 days, the average in-hospital stay lasted for another 22.41 days. With respect to prognostic scores, mean values for qSOFA, SOFA, SAPS 2, and APACHE II were 2.1, 8.7, 63.6, and 23.9, respectively. The 30-day mortality was 22.6% and increased to 29.0% by day 60. Median follow-up period was 998 (95% confidence interval [CI] 467–1529) days. 

Except for arterial hypertension, no sex-specific differences were found in comorbidities, age, BMI, and HIV-status (cf., Appendix A). 

The main cause of admission was infectious or non-infectious pulmonary deterioration (58%) and infectious causes other than pulmonary (11%). Less frequently, intoxication (5%), cardiovascular (8%), neurological (5%), gastrointestinal (5%), or malignant diseases (8%) were causes for inpatient treatment. In total, 22.6% of the patients suffered from a hematologic disease or had been treated for such (Table 1) and 16.1% were diagnosed as a solid tumor before ICU admission, foremost Kaposi’s sarcoma.

Of note, only two of the patients with a solid neoplasia received systemic, cytotoxic chemotherapy. One patient received the last cycle of chemotherapy with pegylated liposomal doxorubicin 105 days before admission to ICU due to metastatic Kaposi’s sarcoma. Another patient suffering from Kaposi’s sarcoma received a final course of doxorubicin 545 days prior to ICU admission. Due to deterioration in general condition, a single patient did not receive treatment for solid neoplasia. All other patients suffering from solid neoplasia underwent surgical treatment before admission to the ICU.

Apart from HIV-1 positivity, there were no infectious complications in a total of 12 patients, while 50 patients suffered from infections other than HIV-1 and/or Hepatitis B/C. Still, the total number of co-infections was slightly positively correlated with HIV viral load (r = 0.462; *p* < 0.001) and slightly negatively correlated with CD4^+^ cell count (r = −0.481; *p* < 0.001). Multiple co-infections were present in 82% of these 50 patients. Typical co-infections are shown in Figure 1. Among bacterial co-infection, *Escherichia coli* (8%), and *Staphylococcus aureus* (5%) were most common, whereas fungal infections were most frequently due to *Pneumocystis jirovecii* (8%), *Candida albicans* (8%), and *Cryptococcus neoformans* (4%). Among viral co-infections, *Cytomegalovirus* (CMV) (12%), Epstein–Barr virus (EBV) (7%), and Herpes simplex virus 1 and 2 (6%) were detected most frequently. In total, *Mycobacterium tuberculosis* was found in five cases, while non-tuberculous mycobacteria were detected in another two patients.

In the studied cohort, only 10 patients revealed HIV-1 and Hepatitis C co-infection. In contrast, Hepatitis B co-infection was more prominent, as 11 patients suffered from an active and/or chronic Hepatitis B and another 19 patients had titers of previous Hepatitis B co-infection.

To investigate the patient outcome, we evaluated median 30-day survival, 60-day survival, and median overall survival (OS), from different aspects. 

Sex was not associated with 30-day, 60-day, or overall survival.

While mean CD4^+^ cell count was significantly lower in those patients deceased before or at 30-day cut-off (i.e., 57.5 ± 72.1 CD4^+^ cells/µL in deceased patients vs. 196.7 ± 276.9 CD4^+^ cells/µL in survivors; *p* = 0.003), before or at 60-day survival (i.e., 66.3 ± 91.1 CD4^+^ cells/µL in deceased patients vs. 206.1 ± 285.4 in survivors; *p* = 0.006) and with regard to overall survival (i.e., 88.2 ± 118.9 in deceased patients vs. 211.4 ± 298.1 in survivors; *p* = 0.029), the area under the curve receiver operating characteristic (AUC-ROC) analysis was not able to determine a significant cut-off value for survival prediction (Appendix A, *p* = 0.099). If split by <50/µL vs. ≥50/µL, as well as <200/µL vs. ≥200/µL, CD4^+^ count univariately did not predict 30-day, 60-day, or overall survival (cf., Table 2).

Likewise, HIV viral load did not sufficiently separate for survival via AUC-ROC analysis (Appendix A, *p* = 0.599). In contrast to CD4^+^ cell count, viral load did not significantly differ between deceased and survivors at 30-day cut-off (i.e., 517 ± 147 thousand HIV copies/µL in deceased patients vs. 572 ± 118 thousand HIV copies/µL in survivors; *p* = 0.901), at 60-day cut-off (i.e., 696 ± 171 thousand copies/µL vs. 501 ± 100 thousand copies/µL, respectively; *p* = 0.657), or regarding OS (i.e., 608 ± 153 thousand copies/µL vs. 529 ± 105 thousand copies/µL, respectively; *p* = 0.415). Still, 23 patients had a viral load < 200 copies/µL. Here, survival was not associated with viral load (i.e., <200 HIV copies/µL vs. ≥200 HIV copies/µL) at 30-day, 60-day, and overall survival (cf., Table 2). Of note, HIV viral load was slightly negatively correlated with CD4^+^ cell count (r = −0.467; *p* < 0.001) and white blood cell count (r = −0.384; *p* = 0.002).

The absence of co-infections was borderline-significant for 30- (*p* = 0.052; Log Rank test) and 60-day survival (*p* = 0.022) but did not significantly subdivide the cohort in terms of OS (*p* = 0.069). In line with HIV viral load, adherent use, incompliant use, or non-existing ART before admission did not alter the outcome at 30 days (*p* = 0.162), 60 days (*p* = 0.481) or in terms of OS (*p* = 0.722). 

However, invasive ventilation (median OS 71 (95% CI 0.0–147.2) days) compared to non-invasive ventilation plus spontaneous breathing (median OS not reached) decreased median OS (*p* = 0.002), while median 30-day (*p* = 0.041) and 60-day survival (*p* = 0.026) was less conclusive but still significant. Moreover, the need for vasopressor treatment negatively impacted 30-day mortality (*p* = 0.008), 60-day mortality (*p* = 0.008), and median OS (vasopressor use: median overall survival 71 (95% CI 0.0–155.2) days, no vasopressor use: median OS not reached; *p* = 0.003). With respect to comorbidities, the presence of hematologic neoplasia did not significantly affect median 30-day (*p* = 0.056), 60-day (*p* = 0.187) or median overall survival (*p* = 0.483). In contrast, solid neoplasms had a significant negative impact on 30-day (*p* = 0.023), 60-day, and overall survival (both *p* < 0.001). 

To gain further insights, we performed subgroup analyses of patients with and without malignant comorbidities (i.e., hematological, and solid neoplasms). These data are presented in Appendix A. Here, patients with malignancies were significantly older (aged 51 years) in comparison to patients without malignant comorbidities (aged 43 years, *p* = 0.018) and CD4^+^ T-cell counts were significantly lower, if a malignant comorbidity was present (*p* = 0.005). However, no significant difference was observed between both subgroups in terms of HIV viral load at admission (*p* = 0.568) and ICU scores (SAPS 2 *p* = 0.523, APACHE II *p* = 0.373, and SOFA *p* = 0.189).

Neither myocardial, rhythmological, nor pulmonal comorbidities (*p* > 0.05 for all comparisons) were associated with survival at the defined time points. Similarly, vascular comorbidities, renal comorbidities, hepatic comorbidities, and neurological comorbidities (*p* > 0.05 for all comparisons) did not affect outcome.

Moreover, Hepatitis C co-infection was not associated with reduced 30-day (*p* = 0.452), 60-day (*p* = 0.772), or overall survival (*p* = 0.953) and, likewise, Hepatitis B co-infection did not confer to inferior outcome (*p* = 0.610) (cf., Appendix A).

Next, we evaluated laboratory parameters and ventilation parameters as cut-off values for median survival using an AUC-ROC analysis (Appendix A). Here, maximum ventilation pressure (Pmax, *p* = 0.009, cut-off value 16.5 cmH_2_O) and positive end expiratory pressure (PEEP, *p* = 0.005, cut-off value 7.5 cmH_2_O), platelet count (*p* = 0.006, cut-off value 164 thousand/µL), international normalized ratio (INR, *p* = 0.025, cut-off value 1.12), and pH level (*p* = 0.004, cut-off value 7.31) were identified as possible separators for median survival. Regarding the detected laboratory parameter cut-offs, HIV copy burden did not correlate with platelet count (r = −0.002; *p* = 0.990), INR level (r = −0.054; *p* = 0.684) or pH-level (r = −0.186; *p* = 0.155). Neither creatinine level (*p* = 0.226), urea level (*p* = 0.272), bilirubin level (*p* = 0.338), body temperature (*p* = 0.240), nor body mass index (*p* = 0.701) had an effect on median survival. However, when considering 30- and 60-day survival, as well as median overall survival, only platelet count and pH-value at admission sustained prognostic parameters via Log Rank and Fisher’s exact tests (Table 2), but ventilation parameters and INR vastly did not (*p* > 0.05).

The introduced organ-failure scoring systems SAPS 2 [25], APACHE II [26], SOFA [27,28,29], and qSOFA [28,30], were likewise analyzed via AUC-ROC to determine cut-off values to separate median overall survival (depicted in Figure 2). Because the qSOFA cut-off value was not statistically significant (*p* = 0.229), we chose the common cut-off value (<2 pts. vs. ≥2 pts.) [28,30]. For SOFA (*p* < 0.001) we identified <7 pts. vs. ≥7 pts. as cut-off value, for APACHE II (*p* < 0.001) it was <25 pts. vs. ≥25 pts. and for SAPS II (*p* < 0.001) < 59 pts. vs. ≥59 pts., respectively.

Except for qSOFA (Log Rank test *p* = 0.326), all other scores performed very well in median overall survival prediction (Table 2). However, when evaluated for 30- and 60- day survival, best survival prediction was achieved using SOFA Score (30-day survival *p* = 0.001, 60-day survival *p* = 0.001), while APACHE II (30-day survival *p* = 0.070, 60-day survival *p* = 0.024), and SAPS 2 (30-day survival *p* = 0.007, 60-day survival *p* = 0.024) were less sensitive.

To test for independence, we performed multivariate survival analyses using the Cox regression model. Those variables that were univariately associated with survival were used to discriminate independence via multivariate analysis. We evaluated the pH level (<7.31 vs. ≥7.31) and platelet count (<164,000/µL vs. ≥164,000/µL), primary ventilation mode (invasive ventilation vs. non-invasive ventilation plus spontaneous breathing) and its specific parameters Pmax (<16.5 cmH_2_O vs. ≥16.5 cmH_2_O) and PEEP (<7.5 cmH_2_O vs. ≥7.5 cmH_2_O), as well as the use of vasopressors during first 24 h of ICU stay (yes vs. no). Moreover, we evaluated the presence and extent of solid neoplasms (no neoplasm vs. localized neoplasm vs. metastasized neoplasm) and the tested organ-failure scores SOFA (<7 pts. Vs. ≥7 pts.), SAPS 2 (<59 pts. vs. ≥59 pts.), and APACHE II (<25 pts. vs. ≥25 pts.) (Table 3). Here, 30-day and 60-day survival were relevantly affected by platelet count and pH-value, but Hazard ratios decreased with increasing survival interval from ICU admission. Apart from this, the presence of a solid neoplasm remained an independent survival predictor regardless of the time point evaluated. Compared to APACHE II and SAPS 2, the SOFA score was a relevant prognostic indicator at multivariate 30- and 60-day survival analysis. However, as it is included in the SOFA score, the platelet count variable excluded the SOFA score variable from the multivariate final survival model by testing for independence. In conclusion, the best median overall survival was predicted by a low APACHE II score, the absence of solid neoplasms and a platelet count > 164,000/µL at ICU admission.

Multivariate survival analyses, including CD4^+^ cell count and HIV viral load, can be found in Appendix A. Inclusion of these factors did not alter the independence of the presented variables and stated the subsequent survival models.

## 4. Discussion

Deriving the average SOFA score, the present cohort is a selected cohort of HIV-1 positive patients in critically ill conditions, from a regionally low incidence area, and treated in the ICU of a German tertiary care university hospital. Compared with non-survivors at 24 h after ICU admission in the original work by Vincent et al., the mean SOFA Score of 8.7 in our cohort is very much alike [29]. Moreover, admission diagnoses in HIV patients treated in the ICU are comparable to global cohort evaluations of Huang et al. [31], and exemplarily compared with data from low incidence areas, such as London in the UK [32], as well as Cape Town in South Africa [33]. Hence, this cohort seems representative enough to evaluate predictors of survival. While some parameters, that are univariately associated with survival in this cohort, are included in ICU risk scores (e.g., platelet count in SOFA score [27], metastatic cancer in SAPS 2 [25], and pH in APACHE II Score [26]), we also examined the independence and influence of multiple other factors and variables in this cohort.

We identified low platelets, low pH value, and the presence of solid neoplasia as relevant risk factors for short-term survival and low platelets, solid neoplasia, and APACHE II score ≥ 25 pts. as risk factors for overall survival. Other variables, such as catecholaminergic treatment, invasive ventilation, high values of PEEP and Pmax univariately negatively associated with outcome but did not sustain a significant risk factor in multivariate analysis. CD4^+^ cell count and HIV viral load at ICU admission, initiation of antiretroviral treatment, the presence or absence of hematological malignancies, and hepatic, renal, cardiac, neurologic, or pulmonal comorbidities had no impact on outcome in this cohort.

### 4.1. Antiretroviral Treatment

In contrast to Neto et al. [34], in our cohort, patient outcome was independent from adherence to ART. In line with our data, Dickson et al. reported comparable outcomes in patients with and without ART [35]. Moreover, a prospective study of early ART initiation in ICU treatment was discontinued due to poor recruitment and similar outcomes in early- and late-onset ART in interim analyses [36]. More evidently, Andrade et al. performed a meta-analysis on ART initiation including 12 studies. Here, ART initiation led to favorable short-term outcomes, but overall survival was not significantly altered after ART initiation [37]. Nevertheless, both immune reconstitution inflammatory syndrome (IRIS) [38] and lactic acidosis due to nucleoside reverse transcriptase inhibitors (NRTI) treatment [39] must be considered as potentially life threatening complications during ICU treatment. Overall, the importance of ART and its specific initiation timing and the correct sequential regimen in ICU treatment of HIV patients remains controversial. Other factors of ART on ICU include the potential for medication interaction via Cytochrome P450 3A4 inhibition, the lack of parenteral regimen and the decreased gastrointestinal uptake in multimorbidity [40]. Therefore, prospective studies are needed to investigate the above-mentioned factors in an ICU setting.

### 4.2. Cancer

Apart from antiretroviral treatment, HIV itself predisposes to the development of malignant diseases. Among hematologic neoplasms, non-Hodgkin lymphomas are, in particular, associated with HIV infection. Most of these lymphomas, such as Burkitt lymphoma, diffuse large B-cell lymphoma, and primary CNS lymphoma, exhibit an aggressive behavior and require intensive immuno-chemotherapy [41]. Interestingly, the incidence of non-Hodgkin lymphoma in the HIV-positive population was significantly decreased with the emergence of ART [42]. Considering this, survival in the present cohort was not reduced due to the occurrence of hematologic neoplasms. In contrast to hematologic neoplasms, solid neoplasms had a significant impact on outcome. However, in patients with solid neoplasms, the need for critical care in the present cohort was not due to subsequent immunosuppressive chemotherapy. Interestingly, the incidence of AIDS-defining Kaposi’s sarcoma decreased during the era of ART, whereas the incidences for, e.g., lung cancer, colorectal cancer, anal cancer, pancreatic cancer, melanoma, and female breast cancer increased [42]. Additionally, recent Surveillance, Epidemiology, and End Results (SEER) database studies have found that HIV-positive patients have fewer localized cancer diagnoses and lower treatment rates compared with the general population [43]. For example, HIV-positive patients are 2.6 times more likely to be diagnosed with melanoma, and furthermore, melanoma demonstrates a more aggressive course in these [44]. We demonstrated that patients with malignant diseases were significantly older at ICU admission and CD4^+^ T-cell count at ICU admission was significantly lower. Yet, ICU risk scores did not sufficiently distinguish between patients with and without a malignant comorbidity (Appendix A). Overall, ICU patients with solid neoplasms are at high mortality risk and further studies are needed to investigate possible socio-economic disparities in diagnosis and treatment, as well as to understand the role of HIV and the effect of ART on cancer progression and treatment.

### 4.3. Platelets

The impact of HIV-1 on megakaryopoiesis and especially platelets [3] remains controversial. It has been shown that the normal lifespan of a platelet of about 9 days is often decreased by ≥50% in HIV-1 positive patients [2,3]. While HIV interacts with platelets as a possible reservoir location [1] in endocytic vesicles on the one hand, in vitro data on the other hand brought forth the idea that platelets might inhibit HIV-replication via secretion of platelet factor 4 [2]. HIV-interaction with platelets may activate primary hemostasis, leading to thrombo-embolic events [2]. These previous factors might explain the critical role of platelet levels in our cohort on overall survival. Here, a platelet count below 164,000/µL was a prognostic parameter estimating reduced short- and long-term survival both univariately and multivariately. Therefore, prognostic organ-failure scores incorporating platelet count, such as SOFA [27], might be used to determine the outcome of HIV-patients in the ICU setting. Unfortunately, we are not able to distinguish between impaired HIV-inhibition or platelet loss due to high viremia and/or pathological activation and disseminated intravasal coagulation. Here, HIV copy load did not significantly differ between patients with <164,000/µL platelets (mean 886,459 ± 1,726,272 copies HIV/µL) and those with ≥164,000/µL platelets (mean 292,110 ± 527,380 copies HIV/µL, *p* = 0.095) by Students’ *t*-test. Hence, the above-mentioned factors should be addressed prospectively in ICU-treated patients with HIV-infection, but we and others [2] suggest a prognostic impact of a lower platelet count on outcome.

### 4.4. Hepatitis B and Hepatitis C Co-Infection

In contrast to Medrano et al. [45], in this patient cohort we were unable to show the negative impact of HIV/Hepatitis C co-infection on outcome in ICU patients. While the Kaplan–Meier curves showed a trend towards reduced survival in Hepatitis C/HIV co-infected patients, the Log Rank test did not reveal statistically significant differences (*p* = 0.953), especially not in a critical short-term ICU-setting. Likewise, Hepatitis B co-infection did not result in significantly reduced overall survival (*p* = 0.610), while Kaplan–Meier curves also indicate a superior long-term survival in non-co-infected patients (cf., Appendix A).

### 4.5. Other Factors and Prognosis Scores

Low pH levels often indicate a critical health status. Bicarbonate loss in renal failure, CO_2_ retention in type II respiratory failure and lactic acidosis due to anaerobic metabolism are most common in ICU-treated patients. In nucleoside reverse transcriptase inhibitor (NRTI)-pretreated patients, pharmacologically induced lactic acidosis can additionally lead to a possible life threatening complication [39,46]. Moreover, acidosis results in reduced response of myocardial and vascular smooth muscle contraction to catecholaminergic therapy [47]. In this cohort, vasopressor use during the first 24 h after ICU admission was univariately predictive for survival. While APACHE II risk score takes the pH value into account [26], SAPS 2 includes bicarbonate [25]. Although acidosis can often be buffered or balanced by hemodialysis, supplementation of sodium bicarbonate, tris-hydroxymethyl aminomethane buffer, or modification of respiratory rate and ventilation pressure within hours, in this cohort, pH value < 7.31 during first 24 h of ICU stay was an independent predictor of reduced 30- and 60-day survival, and APACHE II score was best to predict long-term survival. With respect to ventilatory parameters (PEEP and Pmax), associated with survival univariately. Yet, multivariate survival prediction did not feature PEEP and Pmax as parameters of the final prognostic model, as they might just be an indicator for type II respiratory failure and thus respiratory acidosis. While SAPS 2, APACHE II and SOFA score univariately predicted long-term survival very well, SOFA score was also univariately better to predict short-term survival in our cohort.

### 4.6. General Limitations

Despite the multiparametric approach, the present study shows several limitations. First, the evaluations were performed retrospectively, and patient inclusion criteria can be biased by tertiary care at a single university hospital in an area with low HIV prevalence.

All patients were treated in consultation between specialists in infectiology and intensive care medicine. Although the treatment of patients is based on national and international infectious diseases and intensive care guidelines, a unicentric analysis must face the possibility of systemic errors and biased conclusions. In particular, a single center study always bares the risk of bias due to local routine diagnostics (e.g., blood values measurements, measurements of vital parameters) and treatment procedures (e.g., intravenous fluid management, choice of catecholaminergic agent, availability of antibiotic treatment).

#### 4.6.1. Limitations Due to Local Health Sector Structure

In Münster, the inpatient care of non-critical HIV patients is performed in an infectious disease ward at the university hospital site. Outpatient care is either provided by a resident physician medical practice or by a specialized outpatient unit at the university hospital. Thus, these factors might bias patient allocation, promote a limited or prespecified cohort and, subsequently, bias the retrospective inclusion of locally treated patients.

The provision for hematological care and bone marrow transplantation is one major focus of the hospital, probably overrepresenting hematological malignancies in the present cohort. Moreover, the internal medicine ICU focuses on patients with respiratory and cardiocirculatory failure undergoing ventilation and ECMO treatment, likewise resulting in a possible overrepresentation of these patients in the underlying cohort.

In addition, the total number of evaluated patients is limited to patients treated during the ART era as records of earlier cases provided insufficient data for the parameters evaluated.

#### 4.6.2. Limitations Due to Epidemiology, Education and Provision of ART

While in Germany, HIV prevalence in outpatient care covers about 0.1% of all cases per year [10], the prevalence in South Africa is many folds higher (18.0%) [8], requiring higher volume inpatient and outpatient care.

In addition, for Münster, the proportion of HIV-positive people aware of their infection is about 90%; of these, 96% are compliant with ART treatment. Thus, low prevalence, comprehensive knowledge on the disease and a high level of adherent treatment [48] contribute to a low complication rate, which is reflected in the low number of ICU cases in the evaluated period of time.

Taking the latter aspects into consideration, causes of ICU admission in South Africa during the same period are largely consistent with our data, predominantly requiring respiratory failure treatment (i.e., 72.2% in the South African cohort vs. 58.1% in the present cohort) and treatment of neurological disorders, intoxication, and/or consciousness disorders (i.e., 16.7% in the South African cohort vs. 9.7% in our cohort) [33]. However, the internal medical ICU of Münster university hospital rarely treats post-operative patients. Hence, admission due to post-surgical complications is lower (i.e., 1.6% of the patients in our cohort vs. 11.1% of the patients in the South African cohort).

Compared with a London cohort of HIV-positive ICU patients [32], respiratory failure was less common in that report (i.e., 31.4% lower respiratory tract infections plus 13.0% other infections) but cardiovascular causes (i.e., 3.4%), hematological and oncological causes (i.e., 8.7%), as well as neurological causes (i.e., 9.7%) were comparable with the present cohort [32].

With regard to co-infections, detection of *Mycobacterium tuberculosis* was relevantly less frequent in Germany (5 of 50 patients with infectious complications) than in South Africa (28 of 54 patients), but *Pneumocystis jirovecii* was more common in our cohort (11 of 50 patients with infectious complications) than in the South African cohort of Balkema et al. (6 of 54 patients) [33]. Unfortunately, no data on infectious complications are available from the London cohort.

In conclusion, ICU staff face similar complications in critically ill HIV-positive patients regardless of the epidemiologic setting. Thus, the present data may at least be partially generalizable to other settings. In parallel, the present data harbor risk for previously outlined biases and hence should be interpreted with caution. Taken together, the present findings require replication in larger multicenter cohorts from different parts of the world.

## 5. Conclusions

To our knowledge, we are the first to describe low platelet count as an independent risk factor for survival of HIV patients treated in the ICU. Moreover, we highlighted the need to raise awareness of solid neoplasms as a relevant predictor of survival. Overall, the cohort presents common comorbid diseases and infections. Here, we also demonstrated the practicability and reliability of common organ-failure risk assessment scores, such as SOFA, APACHE II, and SAPS 2, in this subgroup of ICU patients. Further studies are needed to better understand the interaction of megakaryopoietic cells and platelets with the HI-virus in critically ill patients and the impact on coagulopathy and outcome. The role of antiretroviral treatment in an ICU setting and the prognostic impact of solid neoplasms in HIV-positive patients also require further investigation.

## Figures and Tables

**Figure 1 viruses-15-01164-f001:**
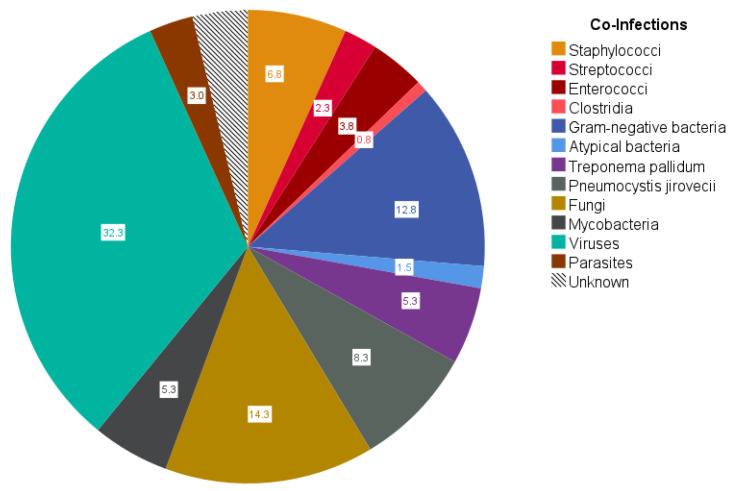
Circle diagram of co-infections or reactivation of infectious diseases in 50 patients with infectious complications at ICU stay. Raw count and frequencies (annotation in the circle diagram) of n = 134 co-infections were documented in 50 patients with infectious causes other than HIV-1 or Hepatitis B and/or C during ICU stay. A relevance weighting was not performed. The category ‘Unknown’ gathers clinically apparent infectious courses without evidence of a specific germ. ‘Staphylococci’: n = 7 *S. aureus*, n = 1 *S. epidermidis*, n = 1 other; ‘Streptococci’: n = 3 *S. pneumoniae*; ‘Enterococci’: n = 4 *E. faecium*, n = 1 *E. faecalis*; ‘Clostridia’: n = 1 *C. difficile*; ‘Gram-negative bacteria’: n = 2 *P. aeruginosa*, n = 10 *E. coli*, n = 3 *K. pneumoniae*, n = 2 *H. influenzae*; ‘Atypical bacteria’: n = 1 Mycoplasma, n = 1 Ureaplasma; ‘*Treponema pallidum*’: n = 7; ‘*Pneumocystis jirovecii*’: n = 11; ‘Fungi’: n = 11 *C. albicans*, n = 2 *C. glabrata*, n = 5 *C. neoformans*, n = 1 Saccharomyces; ‘Mycobacteria’: n = 5 *M. tuberculosis*, n = 2 nontuberculous mycobacteria; ‘Viruses’: n = 1 *Norovirus*, n = 16 CMV, n = 9 EBV, n = 8 HSV-1/-2, n = 4 Influenza A/B, n = 1 VZV, n = 1 Coronavirus, n = 2 Adenovirus, n = 1 *Bocaparvovirus*, n = 1 JC virus; ‘Parasites’: n = 3 *Toxoplasma gondii*, n = 1 *Plasmodium falciparum*; ‘Unknown’: n = 5.

**Figure 2 viruses-15-01164-f002:**
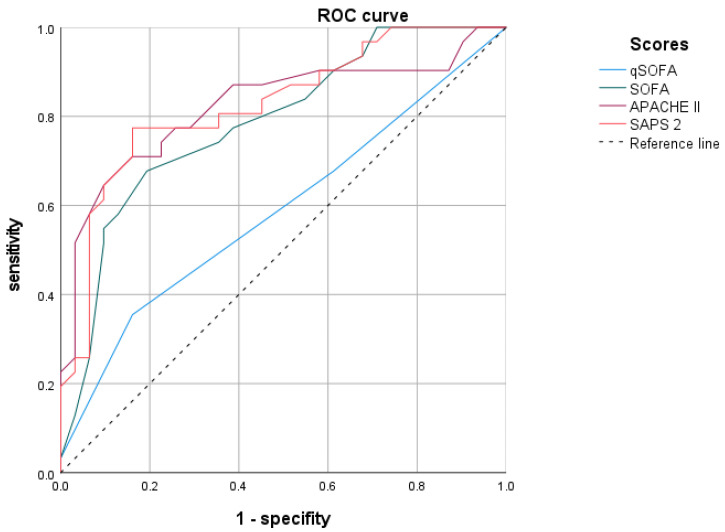
AUC-ROC analysis for median survival regarding qSOFA, SOFA, SAPS 2 and APACHE II. Significance levels of the AUC-ROC analysis are *p* = 0.229 for qSOFA, *p* < 0.001 for SOFA, *p* < 0.001 for SAPS 2, and *p* < 0.001 for APACHE II, respectively.

**Table 1 viruses-15-01164-t001:** Baseline characteristics of the cohort.

Variables		n_total_ = 62	in%
Sex	male	48	77.4
	female	14	22.6
Age (years)	mean (±standard deviation)	46.31 (±11.66)	
Body Mass Index (kg/m^2^)	mean (±standard deviation)	22.28 (±5.34)	
HIV copies (*n* = 60) (thousand copies/µL)	mean (±standard deviation)	559.6 (±124.6)	
CD4^+^ cells (*n* = 60) (cells/µL)	mean (±standard deviation)	164.18 (±251.30)	
Cause of admission	pulmonary cause	36	58.1
	infectious cause (other than pulmonary)	7	11.3
	cardiovascular cause	5	8.1
	gastrointestinal cause	3	4.8
	malignant/immunologic cause	5	8.1
	neurologic cause	3	4.8
	intoxication	3	4.8
HIV initial diagnosis	prior 1 month to ICU stay	37	59.7
	within 1 month to/during ICU stay	25	40.3
HIV type	HIV-1	62	100.0
HIV therapy	compliant use of preexisting ART	28	45.2
	incompliant use of preexisting ART	8	12.9
	newly initiated ART	22	35.5
	ART not initiated	4	6.5
Number of co-infections	0	12	19.4
	1	9	14.5
	2	13	21.0
	3 or more	28	45.2
Hematological neoplasia	Aggressive B-NHL (Burkitt or DLBCL)	7	11.3
	Indolent B-NHL (CLL)	1	1.6
	Hodgkin-Lymphoma	3	4.8
	T-NHL	1	1.6
	MGUS/smoldering myeloma/multiple myeloma	2	3.2
Solid neoplasia	Kaposi’s sarcoma (metast.)	5	8.1
	Lung cancer	1	1.6
	Colon cancer (metast.)	1	1.6
	Urothelial/renal cancer	2	3.2
Ventilation mode (worst of first 24 h at ICU)	spontaneous breathing w/o support	8	12.9
	nasal cannula/face mask	12	19.4
	non-invasive ventilation	16	25.8
	invasive ventilation	26	41.9
qSOFA Score	mean (±standard deviation)	2.08 (±0.82)	
SOFA Score	mean (±standard deviation)	8.73 (±5.16)	
SAPS 2	mean (±standard deviation)	63.55 (±19.00)	
APACHE II Score	mean (±standard deviation)	23.92 (±8.19)	
Overall duration of ICU stay (days)	mean (±standard deviation)	14.48 (±20.06)	
Overall duration of in-hospital stay (days)	mean (±standard deviation)	36.89 (±30.99)	
30-day survival	survived	48	77.4
	deceased	14	22.6
60-day survival	survived	44	71.0
	deceased	18	29.0
Overall survival (days) since ICU admission	mean (±standard deviation) median (95% confidence interval)	2201.53 (±259.04) n.e. *	
Follow-up (days) since ICU admission	mean (±standard deviation) median (95% confidence interval)	1128.32 (±170.05) 998 (467–1529)	

* n.e.: not evaluable, median survival > 50% of the cohort; data on HIV copies as well as CD4^+^ cell count was missing in two cases, hence referring to n = 60 cases; B-NHL: B-cell non-Hodgkin lymphoma; T-NHL: T-cell non-Hodgkin lymphoma; CLL: chronic lymphocytic leukemia; MGUS: monoclonal gammopathy of unknown significance.

**Table 2 viruses-15-01164-t002:** Univariate survival predictors for 30-day, 60-day, and median overall survival (OS).

Variables	30-Day Survival		*p* # *p* *	60-Day Survival		*p* # *p* *	Median OS	*p* #
specifications	n_survived_	n_deceased_		n_survived_	n_deceased_		days (95% CI)	
Sex			# 0.878			# 0.898		# 0.868
Female	11	3	* 1.000	10	4	* 1.000	n.e.	
Male	37	11		34	14		n.e.	
CD4^+^ count (*n* = 60)			# 0.228			# 0.109		# 0.173
<50/µL	21	9	* 0.360	18	12	* 0.158	n.e.	
≥50/µL	25	5		24	6		n.e.	
CD4^+^ count (*n* = 60)			# 0.094			# 0.113		# 0.216
<200/µL	32	13	* 0.155	29	16	* 0.192	n.e.	
≥200/µL	14	1		13	2		n.e.	
HIV viral load (n = 60)			# 0.397			# 0.260		# 0.385
<200 copies/µL	16	7	* 0.356	15	8	* 0.571	n.e.	
≥200 copies/µL	30	7		27	10		n.e.	
pH			# 0.001			# 0.004		# 0.004
<7.31	14	11	* 0.002	13	12	* 0.010	71 (0–160)	
≥7.31	34	3		31	6		n.e.	
Platelet count			# 0.002			# 0.002		# 0.011
<164,000/µL	17	11	* 0.006	15	13	* 0.011	54 (0–303)	
≥164,000/µL	31	3		29	5		n.e.	
Pmax			# 0.036			# 0.031		# 0.016
<16.5 cmH_2_O	29	4	* 0.066	27	6	* 0.055	n.e.	
≥16.5 cmH_2_O	19	10		17	12		96 (20–172)	
PEEP			# 0.037			# 0.017		# 0.002
<7.5 cmH_2_O	35	6	* 0.054	33	8	* 0.037	n.e.	
≥7.5 cmH_2_O	13	8		11	10		71 (6–136)	
INR			# 0.112			# 0.060		# 0.375
<1.12	23	3	* 0.123	22	4	* 0.053	n.e.	
≥1.12	25	11		22	14		n.e.	
qSOFA			# 0.601			# 0.562		# 0.326
<2 pts.	13	3	* 1.000	12	4	* 0.760	n.e.	
≥2 pts.	35	11		32	14		n.e.	
SOFA			# 0.002			# 0.001		# 0.001
<7 pts.	22	0	* 0.001	21	1	* 0.001	n.e.	
≥7 pts.	26	14		23	17		96 (23–168)	
APACHE II			# 0.038			# 0.009		# <0.001
<25 pts.	28	4	* 0.070	27	5	* 0.025	n.e.	
≥25 pts.	20	10		17	13		71 (5–136)	
SAPS 2			# 0.003			# 0.004		# <0.001
<59 pts.	27	2	* 0.007	25	4	* 0.024	n.e.	
≥59 pts.	21	12		19	14		71 (0–152)	

*p* *: Fisher’s exact test; *p* #: Log Rank test; n.e.: not evaluable. pH: negative decadic logarithm of H+, Pmax: maximum ventilation driving pressure, PEEP: positive end expiratory pressure, INR: international normalized ratio, qSOFA: Quick SOFA Screen, SOFA: Sequential Organ Failure Assessment, APACHE II: Acute Physiology and Chronic Health Evaluation II, SAPS 2: Simplified Acute Physiology Score 2.

**Table 3 viruses-15-01164-t003:** Multivariate Cox regression survival analysis.

Variables	30-Day Survival		*p*	60-Day Survival		*p*	Median Overall Survival		*p*
specifications	HR	95% CI		HR	95% CI		HR	95% CI	
pH			0.009			0.013			
<7.31	5.8	1.5–21.9		3.7	1.3–10.4				
≥7.31_(index)_									
Platelet count			0.020			0.013			0.012
<164,000/µL	6.7	1.4–33.7		5.6	1.4–21.4		3.8	1.3–10.6	
≥164,000/µL_(index)_									
Solid Neoplasm			0.026			0.003			0.010
No solid Npl._(index)_									
Localized Npl.	7.9	1.2–52.4	0.031	7.7	1.6–37.5	0.011	5.1	1.5–16.8	0.008
Metastasized Npl.	4.8	0.9–25.3	0.063	6.0	1.5–24.1	0.012	3.0	0.8–11.1	0.110
APACHE II									0.004
<25 pts._(index)_									
≥25 pts.							4.2	1.6–11.1	

*p*-values calculated via forward likelihood ratio test, inclusion criteria 0.05, exclusion criteria 0.10; HR: hazard ratio; 95% CI: 95% confidence interval. Factors evaluated: pH (<7.31 vs. ≥7.31); platelets (<164,000/µL vs. ≥164,000/µL); vasopressor use at first 24 h after ICU admission (yes vs. no); worst ventilation setting at first 24 h after ICU admission (invasive ventilation vs. non-invasive ventilation plus spontaneous breathing); Pmax (<16.5 cmH_2_O vs. ≥16.5 cmH_2_O); PEEP (<7.5 cmH_2_O vs. ≥7.5 cmH_2_O); Solid Neoplasm (No neoplasm vs. localized neoplasm vs. metastasized neoplasm); SOFA (<7 pts. vs. ≥7 pts.); SAPS 2 (<59 pts. vs. ≥59 pts.); APACHE II (<25 pts. vs. ≥25 pts.). Index parameters are marked via (index).

## Data Availability

Upon reasonable request, anonymized data may be obtained from the corresponding authors.

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
