# Peer review of "Risk Factors in HIV-1 Positive Patients on the Intensive Care Unit: A Single Center Experience from a Tertiary Care Hospital"

_viruses, 2023, doi:10.3390/v15051164_

Round 1

Reviewer 1 Report (New Reviewer)

This is a reasonably good clinical paper with important data. There are some corrections that have to be made.

1) There are incorrect technical understanding of statistics throughout the paper. P =< 0.05 shows statistical significance i.e there is some relationship or link but we CANNOT technically use that to show correlations. We cannot claim correlation unless we can show r >= 0.5 or r2 >= 0.25. While the words "correlate"  and "correlation" are used extensively in the paper, r ( correlation coefficient) or r2 ( coefficient of determination) is no where to be found. The authors need to pore through their computed results and use the r or r2 in the respective tables and sentences. Furthermore, the r or r2 are important as it has important implication for the strenght of correlations. Te closer it is to 1.0 the stronger the correlation.

2) What is Mio on Line 25"

3) Typo "HI virus" on line 554

4) There are papers showing that the HIV variant also matters:

https://www.science.org/doi/10.1126/science.abk1688

https://pubmed.ncbi.nlm.nih.gov/31072073/

For example Goh et al suggests the disorder (mutations) at the HIV matrix protein affects the ability of the virus in entering vital organs while disorder (mutations) at the nucleocapsid or capsid could affect the efficiency of the the viral replication and therefore virulence of the virus. Do the authors have such data. If so, it will make their paper even stronger. Interesting paper!!

Author Response

Dear reviewer,

Thank you very much for reviewing our manuscript entitled “Risk factors in HIV-1 positive patients on the Intensive Care Unit: A single center experience from a tertiary care hospital” (initial manuscript ID viruses-2193176, current manuscript ID viruses-2388424). We appreciate the relevant input and subsequently addressed all issues raised in a point-by-point reply.

Reviewer 1

This is a reasonably good clinical paper with important data. There are some corrections that have to be made.

1) There are incorrect technical understanding of statistics throughout the paper. P =< 0.05 shows statistical significance i.e there is some relationship or link but we CANNOT technically use that to show correlations. We cannot claim correlation unless we can show r >= 0.5 or r2 >= 0.25. While the words "correlate"  and "correlation" are used extensively in the paper, r ( correlation coefficient) or r2 ( coefficient of determination) is no where to be found. The authors need to pore through their computed results and use the r or r2 in the respective tables and sentences. Furthermore, the r or r2 are important as it has important implication for the strenght of correlations. The closer it is to 1.0 the stronger the correlation.

We thank the reviewer for this highly relevant input and added the missing information. We chose the correlation coefficient r to represent the correlation. Here we used the r cutoffs presented by Mukaka MM. Statistics corner: A guide to appropriate use of correlation coefficient in medical research. Malawi Med J. 2012 Sep;24(3):69-71. PMID: 23638278; PMCID: PMC3576830. A low positive correlation was determined for r 0.3 to 0.5 and a low negative correlation for r -0.3 to -0.5.

Thus, we changed the passages with correlation analyses into:

[p.8, ll. 300] Still, the total number of co-infections were low positively correlated with HIV viral load (r=0.462; p<0.001) and low negatively correlated with CD4+ cell count (r=-0.481; p<0.001).

and

[p. 9, ll. 355] Of note, HIV viral load was low negatively correlated with CD4+ cell count (r=-0.467; p<0.001) and white blood cell count (r=-0.384; p=0.002).

and

[p. 10, ll. 401] Regarding the detected laboratory parameter cut offs, HIV copy burden did not correlate with platelet count (r=-0.002; p=0.990), INR level (r=-0.054; p=0.684) or pH-level (r=-0.186; p=0.155, data not shown).

2) What is Mio on Line 25"

We thank the reviewer for this input. This must have been corrected by the German language spelling program in Microsoft Word. We double checked and changed all subsequent “mio.” to million.

3) Typo "HI virus" on line 554

We thank the reviewer for this correction. We changed this into “HIV”.

4) There are papers showing that the HIV variant also matters:

https://www.science.org/doi/10.1126/science.abk1688

https://pubmed.ncbi.nlm.nih.gov/31072073/

For example Goh et al suggests the disorder (mutations) at the HIV matrix protein affects the ability of the virus in entering vital organs while disorder (mutations) at the nucleocapsid or capsid could affect the efficiency of the viral replication and therefore virulence of the virus. Do the authors have such data. If so, it will make their paper even stronger. Interesting paper!

We thank the reviewer for this highly relevant input. However, blood samples are no more attainable, as prospective biobanking in an intensive care unit setting is critical due to ethical concerns regarding patients consent. Hence, we cannot provide any data on specific HIV variants and mutations. Moreover, the treatment-relevant genotype analysis was not communicated via the hospital information system due to data protection concerns in patients with HIV, which overall restricts clinical research in this specific subset.

We hope that we fully addressed the reviewers’ issues and the paper is now more suitable for publication in MDPI viruses.

Kind regards from Münster, Germany

Reviewer 2 Report (New Reviewer)

Schulze et al. report on clinical courses and their determinants of HIV-patients on intensive care units. The data are interesting, however, I strongly recommend the authors to remove one statement. In lines 256-258, the authors explain that adjustment of the required significance level to multiple testing was not performed due to the explorative design of their study. This makes no sense, because in case of explorative studies without an evidence-based hypothesis, it is even more likely that some of the measured associations are just by chance. I understand that the authors avoided corrections for multiple testing, because otherwise much higher case numbers would have been necessary to indicate minor differences. Such practical considerations make sense and are thus difficult to reject, but the authors should abstain from explaining their decision with lacking epistemiological need, because this is simply not the case.

Author Response

We thank the reviewer for this highly relevant input and changed this sentence into: “An adjustment to multiple testing was not determined.”

We hope that we fully addressed the reviewers’ issues and the paper is now more suitable for publication in MDPI viruses.

This manuscript is a resubmission of an earlier submission. The following is a list of the peer review reports and author responses from that submission.

Round 1

Reviewer 1 Report

Thank you for the opportunity to review the manuscript titled "Risk factors in HIV-1 positive patients on the Intensive Care Unit: A single center experience from a tertiary care hospital".

Authors investigated the health profiles of HIV-positive patients treated on non-operative ICU between 2009 and 2019 in Germany.

Robust and clear methods were used for data creation and analyses. The results provided significant insight into the prognostic factors and survival among such vulnerable population.

Overall, the study is well organized and is expected to have a significant contribution to our knowledge on the health of people aging with HIV.

Below are some areas that can be improved:

1-     Contextual information needs to be expanded. Especially issues related to AIDS current profile and health services in Germany. This can help researchers compare and contrast results with other populations.

2-     How would authors compare their sample to the general population of AIDS patients in Germany and globally?

3-      What do authors mean by ‘our’ ICU?

4-     How were missing and outlier data managed?

5-     The study setting might have something to do with the findings. Given all patients were treated on same unit, to what extent this might have caused systematic errors and biased conclusions? Can authors say more about how the care in this setting is comparable to the national and international standards? I understand that this might not be an easy task given there is a clear conflict of interest with all authors being affiliated with this same setting. However, this issue needs to be well clarified. Acknowledging it as just a limitation would not clear the concern.

 Thank you!

Reviewer 2 Report

Schulze et al. conducted a retrospective administrative database based single center study of risk factors for survival in 62 HIV patients treated in a single medical ICU during the period 2009-2019. 

The authors analysed an array of parameters as predictors of 30-day and 60-day survival. Unfortunately, the small sample size and the heterogeneity of the study population reduce substantially the confidence to the study findings and the generalizability.

Round 2

Reviewer 2 Report

Unfortunately, in my view the authors have not addressed adequately the issues raised. Hence, I am sorry to recommend to reject the manuscript.

Author Response

Dear reviewer,

thanks again for your highly relevant input.
We are sorry, that we did not meet your expectations and in your eyes were not able to emphasize the generalizability of our data. However - as depicted in the revised manuscript - the small sample size derives from the setting in a Western European country with low incidence and prevalence. Moreover - as elaborated in the revised manuscript - an ICU in South Africa faces the same problems in treatment of HIV-infected patients as one in Great Britain or a rural area of Germany. Hence, we think that our publication is of interest and value for the proposed sepcial issue in Viruses MDPI.

Kind regards from Münster, Germany

Arik Schulze

(on behalf of all authors)